# Reclaim and Valorization of Sea Buckthorn (*Hippophae rhamnoides*) By-Product: Antioxidant Activity and Chemical Characterization

**DOI:** 10.3390/foods11030462

**Published:** 2022-02-04

**Authors:** Cristina Mihaela Luntraru, Livia Apostol, Oana Bianca Oprea, Mihaela Neagu, Adriana Florina Popescu, Justinian Andrei Tomescu, Mihaela Mulțescu, Iulia Elena Susman, Liviu Gaceu

**Affiliations:** 1Hofigal Export Import S.A., Research Development Patents Department, No. 2 Intrarea Serelor Street, District 4, 042124 Bucharest, Romania; cristina.luntraru@hofigal.eu (C.M.L.); mihhaela_neagu@yahoo.com (M.N.); adrianaflorinabira@yahoo.com (A.F.P.); andrei.tomescu@hofigal.eu (J.A.T.); 2National Research & Development Institute for Food Bioresources-IBA Bucharest, 6 Dinu Vintila St., 0211202 Bucharest, Romania; mihaela.multescu@gmail.com (M.M.); iulia.susman@bioresurse.ro (I.E.S.); 3Faculty of Food and Tourism, Transilvania University of Brasov, 29 Eroilor Blvd., 500036 Brasov, Romania; gaceul@unitbv.ro; 4CSCBAS &CE-MONT Centre/INCE-Romanian Academy, 010071 Bucharest, Romania; 5Assoc. m. Academy of Romanian Scientists, 030167 Bucharest, Romania

**Keywords:** sea buckthorn pomace, ultrasound extraction, maceration, antioxidant capacity, biochemical characterization

## Abstract

The by-product resulting from the production of the sea-buckthorn (*Hippophae rhamnoides*) juice may be a functional food ingredient, being a valuable source of bioactive compounds, such as polyphenols, flavonoids, minerals, and fatty acids. For checking this hypothesis, two extracts were obtained by two different methods using 50% ethyl alcohol solvent, namely through maceration–recirculation (E-SBM) and through ultrasound extraction (E-SBUS), followed by concentration. Next, sea-buckthorn waste (SB sample), extracts (E-SBM and E-SBUS samples) and the residues obtained from the extractions (R-SBM and R-SBUS samples) were characterized for the total polyphenols, flavonoid content, antioxidant capacity, mineral contents, and fatty acids profile. The results show that polyphenols and flavonoids were extracted better by the ultrasound process than the other methods. Additionally, the antioxidant activity of the E-SBUS sample was 91% higher (expressed in Trolox equivalents) and approximately 45% higher (expressed in Fe^2+^ equivalents) than that of the E-SBM sample. Regarding the extraction of minerals, it was found that both concentrated extracts had almost 25% of the RDI value of K and Mg, and also that the content of Zn, Mn, and Fe is significant. Additionally, it was found that the residues (R-SBM and R-SBUS) contain important quantities of Zn, Cu, Mn, Ca, and Fe. The general conclusion is that using the ultrasound extraction method, followed by a process of concentrating the extract, a superior recovery of sea-buckthorn by-product resulting from the juice extraction can be achieved.

## 1. Introduction

Globally, along the entire agri-food supply chain, enormous amounts of wastes and by-products are generated. In the context of world population growth and a future crisis in agri-food resources, capitalizing on the by-products of the food industry is one of the challenges of today, attracting the attention of both researchers and producers in the agri-food sector [1,2,3,4,5].

*Hippophae rhamnoides*, also known as sea-buckthorn or sea berry is a species of plant in the *Elaeagnaceae* family, native to the cold-temperate regions of Europe and Asia, especially China, which has large amounts of these crops [6,7,8,9]. The plant is used in the food and cosmetics industries, in traditional medicine, as animal fodder, in horticulture, and for ecological purposes [10].

Due to the high content of vitamin C [11] and high oil content (about 10% of the fresh fruit weight) rich in polyunsaturated fatty acids n-3, n-6, and n-9, sea buckthorn fruits are used in food industry products (juice, drink, smoothie, jam, sauce, oil) and alcohols (wine, liqueur, beer additive) [12,13].

Sea buckthorn berries are also attractive due to their nutritional and health-promoting properties that are known from traditional medicine (inflammation reduction properties, anti-toxic effect, a positive effect on the regeneration and condition of hair and skin) [14] and have been well recognized scientifically for being rich in vitamins, carotenoids, flavonoids, sterols, and tocopherols [8,12,15].

Numerous studies have shown that waste from fruit processing has a high content of bioactive compounds with high antioxidant capacity [16,17], with a beneficial role on human health [18,19,20]. International scientific research has focused on the characterization of fruit by-products and their further incorporation for the development of functional foods and beverages [21]. Thus, research has been done into obtaining concentrated ingredients in biocompounds, such as grape pomace [22,23], apple pomace [24,25], avocado peel and seeds [26], banana peel, tomato skin and pomace, carrot pomace, cauliflower stems and leaves [27,28,29,30], and potato peels [31,32].

Choosing the most appropriate extraction method for obtaining bioactive compounds from agri-food by-products is a broad topic of discussion. On the one hand, we are looking for ways to increase the amount of valuable nutrients and biocomposites while optimizing the extraction parameters in order to minimize the negative impact on the obtained extract, and on the other hand, we aim to reduce costs and processing time. Additionally, a constant challenge is finding the best method to ensure a low impact on the environment and consumer health, reduce energy costs, and use safe solvents. The chosen extraction method is characterized by parameters such as the type of solvent, its pH, the solid–liquid ratio, the process temperature, and the contact area between the solid and the solvent. In turn, these variables affect the energy consumption, the quantity of solvent used and its recovery capacity, the extraction yield, and other factors, which have been increasingly studied through the optimization of parameters and comparison between techniques for different target compounds and their matrix [33,34,35].

Currently, in both industry and research, novel extraction methods such as ultrasound-assisted extraction (UAE) are being proposed for testing and application as alternatives to classical extraction technologies [36]. This extraction process is more efficient and less harmful to the environment, has a shorter extraction time, uses less solvent and energy consumption [37].

Several studies have shown that the use of high ultrasound power, corresponding to high frequencies 20–25 kHz, lead to the fragmentation of the liquid in which they are propagated by forming areas of high and low pressure. This phenomenon is defined as cavities, more precisely gas micro bubbles. A closer examination showed that the application of ultrasound on a plant matrix does not work through an independent mechanism but a combination of fragmentation, erosion, capillarity, detexturation, and sonoporation, which lead to an extraction yield increase [38].

Nikita Sawal et. al. (2021) [39] describes in her study a method for extracting oils from sea-buckthorn seed using UAE. It was concluded that this process has a significant impact on the quality and efficiency of extraction, one of the aspects being the short time of extraction of 8.53 min, and a low plant-solvent ratio of 1:10, obtaining values of 4.1 µg mL^−1^ for antioxidant activity by DPPH method, 181.6 mg/100 g β-carotene eq., and 2.7 g·hg^−1^ oleic acid [39].

Gabriela Isopescu et al. (2019) [40] tried to find the optimum UAE extraction parameters of oils from sea-buckthorn seeds by applying response surface methodology. The key parameters used in the design of the experiment were ultrasonic intensity, temperature, and time of exposure to ultrasound. The best efficiency (87.4 yield) was determined using an ultrasound intensity of 13.77 W/cm^2^, 40 °C, and an exposure time of 10 min [40].

Maceration is a popular and inexpensive technique used for the extraction of oils and active compounds from plant materials. In this process, the plant is grinded and mixed with the solvent and kept in a closed vessel for a minimum of 3 days. It is usually a repeated extraction in order to increase the extraction yield [41].

During maceration, the plant sample is kept in contact with a solvent until the soluble matter is dissolved into the solvent at room temperature. This process softens and breaks plant cell walls in order to release the soluble phyto-chemicals [42].

Several parameters may influence the yield of extraction including: extraction time, temperature, solvent-to-sample-ratio, solvent type and polarity, and the number of repeat extractions of the sample [43,44]. These differences could be attributed to the different properties of active principles in plant [45].

Aim of this study was to reclaim the waste material resulting from the production of sea buckthorn juice and to evaluate the best extraction method between maceration and UAE to obtain the highest yield of nutrients and bioactives from such by-product. To this purpose, both the extracts and the residues resulting from the extraction of seaberry by-product were evaluated in terms of total polyphenols, flavonoids, antioxidant activity, mineral content, total lipid content, and fatty acid composition. The study of the nutritional potential of sea buckthorn by-product may lead to new opportunities for obtaining nutraceuticals and natural functional food at low prices. Effective valorization of agri-food industrial wastes/by-products is envisaged to contribute toward an improved economy as well as minimize the negative impacts on the environment, with positive effects on ensuring food security [1,35].

## 2. Materials and Methods

### 2.1. Plant Material and By-Product Production

The processed sea buckthorn (*Hippophae rhamnoides*) fruits were a mix of 4 varieties of organically certified sea buckthorn from a plantation in Gorj County, Romania.

The fruits were harvested in August 2021, quickly frozen at −44 °C, then stored for 3 months at −25 °C until processing. The fruits were sorted, cleaned and cold pressed in order to obtain the juice. The obtained by-products, that is, the skin and seeds, were dried at a temperature of 40–45 °C, up to a humidity of 10%, for about 70 h.

About 63 kg of frozen sea buckthorn fruit were processed, obtaining 6.3 kg of by-product. One kilogram of dry by-product was used to obtain a concentrated extract sample. One hundred grams of dry by-product was used for the chemical characterization of the product.

The by-product of sea-buckthorn fruits resulting from the juice extraction, dried, and grinded (SB) was purchased from a local supplier from Bucharest, Romania.

### 2.2. By-Product Extraction

Two different extraction methods were used to highlight the performance and yield limits in the extraction of functional biocompounds. Extraction by maceration is done without introducing energy into the system, which that can be an important advantage in the production of large quantities. The obvious disadvantage is the long extraction time [46]. The ultrasound extraction method is a modern method that has the advantage of a shorter process time, but it requires the introduction of energy into the system. This method has gained particular attention due to its use of low-cost equipment, simplicity, and a higher efficiency compared to solvent extraction because of reduced heat and solvent expense [47,48].

The extractions were performed with in two processes:

Seven-day maceration with five recirculations and a final leaching. A 1000 g sample of sea-buckthorn by-product was extracted with 3000 mL ethanol 50% (E-SBM sample);(a)Ultrasonication in an extraction tank connected with an ultrasonic generator (STEEL^®^ Ultrasonic Generator 500-DG), with a frequency of 38 kHz, amplitude of 2.5 µm, power 100%. Extraction time was 10 min, and temperature increased from 25 °C (initially) to 70 °C at the end of the process. A 1000 g sample of sea-buckthorn by-product was extracted with 3000 mL ethanol 50% (E-SBUS sample).

The resulting extracts were filtered through silk cloth and concentrated in a rotavapor (Buchi R-300) connected with a recirculating chiller (Buchi F-308) to a dry weight of approximately 80% (82.75% for E-SBM and 76.43% for E-SBUS) at a maximum temperature of 50 °C and stored in tightly closed containers at 4 °C.

The residual by-product from each extraction process, R-SBM and R-SBUS, was dried at 40 °C and further analyzed (Table 1).

### 2.3. Chemicals and Reagents

The chemical analysis were realized using the following chemical solutions and reagents:−96.9% Pharmaceutical Ethanol for by-product extraction;−Honeywell-Ethanol, Puriss. p.a., ACS reagent, Reag. Ph. Eur., 96% (*v/v*), 2,2,4-Trimethylpentane Reagent Grade, ≥99%, Sodium Sulfate Puriss. p.a., ACS Reagent, Reag. ISO, Reag. Ph. Eur., anhydrous, ≥99.0%;−Sigma Aldrich-Gallic Acid Reference, Sodium Carbonate anhydrous;−Merck-2,4,6-Tri(2-pyridl)-1,3,5-triazine, Palmitic acid Certified Reference, Methyl Palmitoleate Certified Reference, Oleic acid Certified Reference, Linoleic Acid Certified Reference, Linolenic acid Certified Reference;−VWR Chemicals—Folin–Ciocalteu’s Reagent, Sodium Acetate trihydrate, ACS/Reag. Ph.Eur, Sodium Nitrite, Sodium Hydroxide, Ferrous Sulfate heptaydrate (ACS), Hydrochloric acid 37% Ph. Eur., Acetic Acid glacial (ACS/Reag. Ph.Eur), Hydrochloric Acid 32% Ultrapure NORMATOM^®^, Ultrapure for trace metal analysis, Nitric Acid 67%, NORMATOM^®^, Ultrapure for trace metal analysis, Zn, Cu, Mn, Ca, Fe, Mg, Na, K Standard solutions 1000 mg/L, AAS, Petroleum spirit, 40–60 °C, AnalaR NORMAPUR^®^ ACS analytical reagent (max. 0.01% aromatic hydrocarbons), Methanol ≥ 99.8%, HiPerSolv CHROMANORM^®^ Reag. Ph. Eur., gradient grade for HPLC, n-Hexane ≥ 99%, PESTINORM^®^ Supra Trace for organic trace analysis;−Alfa Aesar-Aluminum Chloride anhydrous, Ammonium Acetate, Iron (III) Chloride anhydrous;−PhytoLab-Rutin Reference;−Cayman Chemical-Quercetin Reference;−Panreac-Copper (II) Sulfate pentahydrate (pure Ph. Eur., USP);−Acros Organis-Trolox Reference, Neocuproine hemihydrate.

All the results for the concentrated extracts were calculated for the dry weight.

### 2.4. Determination of Total Polyphenols (TPC)

The total polyphenols content (TPC) was evaluated using Folin–Ciocâlteu method [49]. SB, E-SBM, E-SBUS, R-SBM, and R-SBUS samples were extracted with ethanol 50% under reflux, and the concentrated extracts were diluted with ethanol 50%. Sample extracts were mixed with 5 mL of 10% Folin–Ciocâlteu’s phenol reagent and let to stay for 3 to 5 min at room temperature. The extraction volumes used were 0.05 mL for SB, R-SBM, and R-SBUS, 0.015 mL for E-SBM, and 0.03 mL for E-SBUS. After incubation, 4 mL of 7.5% Na_2_CO_3_ was added, and the reaction mixture was mixed thoroughly and left to stand for 1 h at room temperature. The absorbance was measured at 765 nm against water, with a Jasco V-530UV-VIS spectrophotometer, and total polyphenolic content was calculated using a Gallic acid standard calibration curve with concentrations ranging from 1 to 5 μg/mL and expressed as Gallic acid equivalent per gram.

### 2.5. Determination of Total Flavonoid (TFC) as Rutin Equivalent

TFC was evaluated using the aluminum chloride colorimetric method as described in the Romanian Pharmacopoeia (2020) [50]. The sea-buckthorn by-product was extracted with ethanol 50% under reflux, and the concentrated extracts were diluted with ethanol 50%. Sample extracts were mixed with 5 mL of 1.219 mol/L sodium acetate, 3 mL of 0.187 mol/L aluminum chloride, and 12 mL methanol. The extraction volumes used were 5 mL for SB, R-SBM, R-SBUS, and E-SBM and 4 mL for E-SBUS. The reaction mixture was mixed thoroughly and incubated for 20 min at room temperature. The absorbance was measured at 430 nm against a blank prepared with 5 mL diluted extract and 20 mL methanol, with a Jasco V-530 UV-VIS spectrophotometer, and TFC was calculated using a rutin standard calibration curve with concentrations ranging from 20 to 90 µg/mL and expressed as rutin equivalent per gram.

### 2.6. Determination of Total Flavonoid (TFC) as Quercetin Equivalent

In this method TFC was measured as quercetin equivalent using the aluminum chloride colorimetric method as described by Turturica, M. et al. (2015) [51]. The sea-buckthorn by-products were extracted with ethanol 50% under reflux and the concentrated extracts were diluted with ethanol 50%. Sample extracts were mixed with 4 mL distilled water and 0.3 mL of 5% sodium nitrite then let to stay for 5 min at room temperature. A 0.3 mL sample of 10% aluminum chloride was added and the mixture was left to stay another 6 min at room temperature. The extraction volumes used were: 0.5 mL for SB, R-SBM, R-SBUS, and E-SBM and 0.35 mL for E-SBUS. After this, 2 mL of 1 mol/L sodium hydroxide was added, and the volumes were brought up to 10 mL with distilled water. The absorbance was measured after 15 min at 510 nm against a blank prepared with water instead of sample extract, with a Jasco V-530 UV-VIS spectrophotometer, and TFC was calculated using a quercetin standard calibration curve with concentrations ranging from 5 to 100 μg/mL and expressed as quercetin equivalent per gram.

### 2.7. Antioxidant Activity CUPRAC Assay

The antioxidant activity was measured using the CUPRAC assay, as described by Apak et al. (2006) [52]. The sea-buckthorn by-product was extracted with ethanol 50% under reflux, and the concentrated extracts were diluted with ethanol 50%. One milliliter of 10^−2^ mol/L copper sulphate was mixed with 1 mL of 7.5 × 10^−3^ mol/L neocuproine, 1 mL of 1 mol/L ammonium acetate buffer with pH 7.0, 0.1 mL sample extract, and 1 mL of water. The reaction mixture was mixed and incubated for 30 min at room temperature. The absorbance was measured at 450 nm against a blank sample prepared with water instead of sample extract, with a Jasco V-530 UV-VIS spectrophotometer, and the antioxidant activity was calculated using a Trolox standard calibration curve with concentration ranging from 10 to 60 μg/mL and expressed as Trolox equivalent per gram.

### 2.8. Antioxidant Activity FRAP II Assay

The antioxidant activity was measured using the FRAP II assay, as described by Szydłowska-Czerniak et al. (2012) [53]. The sea-buckthorn by-product was extracted with ethanol 50% under reflux, and the concentrated extracts were diluted with ethanol 50%. Sample extracts were mixed with 2 mL of freshly prepared FRAP II reagent (2.5 mL of a 10 mmol/L TPTZ (2, 4, 6-tripyridyl-s-triazine) solution in 40 mmol/l HCl, 2.5 mL of 20 mmol/L FeCl_3_, and 25 mL of 0.1 mol/L acetate buffer, pH 3.6, and incubated at 37 °C for 10 min), and the volumes were brought up to 10 mL with distilled water. The extraction volumes used were 0.015 mL for SB; 0.02 mL for E-SBM, E-SBUS, and R-SBUS; and 0.025 mL for the R-SBM sample. The absorbance was measured after 10 min at 593 nm against a blank prepared with water instead of sample extract, with a Jasco V-530 UV-VIS spectrophotometer, and the antioxidant activity was calculated using a FeSO_4_·7H_2_O standard calibration curve with concentrations ranging from 1 to 10 µg/mL and expressed as Fe^2+^ equivalent.

### 2.9. Mineral Content Analysis

Mineral content was determined using an atomic absorption spectrophotometer (ContrAA 700; Analitykjena) [54]. Total ash was determined by incineration at 650 °C in an oven. Analyses were performed using an external standard and calibration curve for all minerals and were obtained using 6 different concentrations. Dried samples were digested in mixture of concentrated HNO_3_ and HCl, one part of the former to three parts of the latter by volume.

### 2.10. Total Lipid Content

Lipid extraction was performed in a continuous extraction apparatus (Soxhlet -VELP). Five grams of the sample was weighted in a cellulose fingerstall, and then 50 mL of n-Hexane was added. The specific program of the equipment was used, and after solvent recovery the extracted lipidic mass was weighted [55].

### 2.11. Fatty Acid Composition

The analysis was performed on a Trace 1310 gas chromatograph, fitted with a capillary column (TG-WAXMS, 30 m × 0.25 I.D., 0.25 µm film thickness), and equipped with a ISQ 7000 MS detector, from Thermo Scientific, combined with a Chromeleon 7 Cromatography data system. Helium was used as a gas carrier with a flow rate of 1.5 mL/min under constant pressure. Analysis was conducted in split mode with an injection volume of 1 µL. The GC oven was programmed as follows: 160 °C for 5 min, 160 °C to 200 °C at a rate of 4 °C/min, held for 4 min, then raised again to 250 °C at a rate of 10 °C/min and held for 2 min. The inlet was kept at 250 °C, and the MS detector temperatures for transfer line and ion source were 250 °C, and the recording mass spectra range was set between 50–500 *m/z*.

Qualitative analysis was achieved through comparison (retention time and ion fragments) of the mass spectrum data of each fatty acid in the sample with the mass spectra obtained from NIST Mass Spectral Library.

The quantification of fatty acids was performed using a calibration curve obtained by injecting a solution of 0.05, 0.1, 0.2, 0.3, and 0.4 mg/mL of Palmitic acid, Methyl Palmitoleate, Oleic acid, Linoleic Acid, and Linolenic acid reference standards.

The fatty acids quantification was performed on the lipids extracted with the Soxhlet apparatus. 0.1 mg of the extracted lipids were dissolved in 0.5 mL of Petroleum Ether and 9.5 mL of a solution of 0.5 mol/L HCL in MeOH, then boiled at 65 °C until the solution was clear and then boiled for another 5 min. The solutions were cooled and transferred into a separation funnel. Twenty milliliters of isooctane was added. The isooctane layer was washed with distilled water up to neutral pH. The obtained solutions were dried from possible existing water droplets by mixing them with anhydrous sodium sulfate powder and then filtered through glass microfiber filters (Whatman CAT No. 1822-070) and diluted 1:5, and 1 µL of each was injected into the GC-MS system [54].

### 2.12. Statistical Analysis

All analyses were executed in triplicate and the mean values with the standard deviations were related. Microsoft Excel was used for statistical analysis, with the level of significance set at 95%. Analysis of variance (ANOVA) and Tukey’s test was used to estimate statistical differences between samples. Differences were considered significant for a value at *p* < 0.05. For all 5 determinations, the analyses were done in triplicate and the average value and standard deviation were calculated. Pearson correlation coefficients were calculated using SPSS 23.0 (SPSS Inc., Chicago, IL, USA).

## 3. Results and Discussions

### 3.1. Total Polyphenol Content, Flavonoid Content, and Antioxidant Activities 

Table 2, Table 3, Table 4 and Table 5 Report the Total Polyphenols (TPC Expressed in mg GAE/g); Total Flavonoid Content (TFC Expressed in mg RE/g and mg QE/g); and Antioxidant Activity (Expressed in mg Trolox/g and mg/g Fe^2+^) Content of SB, R-SBM, R-SBUS, E-SBM, and E-SBUS Samples.

From Table 2, it can be seen that from both maceration and ultrasound processes, the residues R-SBM and R-SBUS have similar TPC results. Comparing the results obtained for the extract samples, E-SBUS has a polyphenol content approximately 53% higher than E-SBM. It can also be observed that E-SBM and E-SBUS have a much higher content of polyphenols than SB, 4.5 times higher than R-SBM, and 7.1 times higher than R-SBUS. Similar TPC concentrations were reported in sea buckthorn by Bittová et al. (2014) [56]. These results show that by advanced processing of by-products, that is, by various extraction methods, compounds with a high content of polyphenols can be obtained, which can be used to acquire nutraceuticals or various food supplements. In the literature, research has been undertaken on TPCs on different varieties of sea buckthorn, with wide ranges of results. The total phenolic content of the berries studied by Ercisli et al. (2007) [57] ranged from 21.31 mg GAE/g on a dry-weight basis to 55.38 mg GAE/g. On the other hand, Guo et al. (2017) [58] studied four varieties of sea buckthorn from China and determined values for the total content of polyphenols averaging only 38.7 mg GAE/g (DW).

Regarding the flavonoid content (Table 3), it can be seen that there are no significant differences among SB, R-SBM, and R-SBUS samples, expressed in both rutin and quercetin equivalents.

When comparing the concentrated extracts, it can be found that the ultrasound extract E-SBUS has a much higher content of flavonoids, with a significant difference of 59% for quercetin. Regarding the total flavonoid content expressed in rutin, there are no significant differences between E-SBUS and E-SBM. Similar results regarding the TFC value expressed in mg RE/g were reported by Li, Y. et al. (2021) [59], who studied five varieties of sea berry. Regarding the TFC expressed in mg QE/g, similar results were presented by Criste A. et al. (2020) [60], who studied four types of Romanian sea buckthorn berries.

Additionally, as in the case of polyphenols, it can be seen that the rutin expressed flavonoid content of E-SBM and E-SBUS is 3 times higher than SB. The TFC expressed in quercetin of the E-SBM sample was 2.5 times higher than the SB sample. The TFC expressed in QE of the E-SBUS sample was 4 times higher than in the case of the SB sample.

Regarding the antioxidant activity, CUPRAC assay is used to measure the antioxidant capacity of plants and the cupric ion reducing ability of polyphenols, vitamins C, and vitamin E [32], while the FRAP II method is sensitive in the measurement of total antioxidant power of plants and pharmacological plant products [33].

The antioxidant activity (Table 4) measured by both methods shows similar results to those obtained for polyphenols. It can be seen that ultrasound extraction is more efficient than maceration, with the antioxidant activity being significantly higher for E-SBUS samples compared to E-SBM (91% higher expressed in TE and 45% higher expressed in Fe^2+^ equivalents). Additionally, the values of the antioxidant activity obtained for the E-SBM and E-SBUS samples are significantly higher than those of the SB sample (4 and 2.5 times higher, respectively, by the CUPRAC method; 4.1 and 2.8 times higher, respectively, by the FRAP II method). Among SB, R-SBM, and R-SBUS, there are no significant differences regarding antioxidant activity, expressed in both CUPRAC and FRAP II methods. Similar results were obtained by Perk A.A. et al. (2016) [61] and Korekar G. et al. (2014) [62] regarding the antioxidant activity using CUPRAC and FRAP II methods.

### 3.2. Mineral Content Analysis

From the mineral analysis, it can be seen that the extraction processes gave different results for each studied mineral, so a general conclusion cannot be drawn. E-SBUS gave better results than E-SBM for the extraction of calcium (by 20.38%), iron (by 15.04%), sodium (by 11.7%), and potassium (by 19.2%). In the case of magnesium, E-SBM was more efficient by 38.8% compared to E-SBUS. No significant differences were found among zinc, copper, and manganese contents (Table 5).

Thus, in 100 g of the two extracts (E-SBM and E-SBUS), more than 25% of the Daily Values are found (RDI-FDA 2011) for potassium. Magnesium content ws 50% of RDI (Table 6) in 100 g of extract obtained by maceration and 45.75% of RDI in the extract obtained by the ultrasound process. Additionally, the two extracts have contents of manganese, zinc, and iron that cannot be neglected.

Data concerning sea buckthorn samples on the content of minerals in the literature differ greatly from several points of view, such as variety or species, climate, composition of soil, part of the plant that is studied, area of cultivation, degree of maturity, etc., elements that are normal. All these factors also manifest themselves in the case of sea buckthorn samples of different origin [8,64].

### 3.3. Total Lipid Content

Table 7 shows the total lipid content determined for the five studied samples.

These results show that the quantities of lipids in the concentrated extracts are extremely low, compared with the content from SB and R-SBM and R-SBUS. Dulf et al. (2012) [65] reported the oil content of whole fruit, pulp, and seeds (taking into account the fresh weight) of different sea buckthorn varieties in Romania: 45–84 g/kg in whole fruit, 45–88 g/kg in pulp, and 106–135 g/kg in seeds.

A known aspect is that ethanol solutions are not a solvent with the ability to solubilize and extract fatty substances from the vegetable matrix. Additionally, the results showed that even the effect of the cavities produced by ultrasound do not facilitate the transfer of this compounds to the solvent.

The chromatographic profile of fatty acids (Figure 1) shows the presence of myristic acid, palmitic, methyl palmitoleate, methyl isostearate, oleic acid, methyl 10-octadecenoate, linoleic, and linolenic in all the samples analyzed.

Using the calibration curve, five fatty acids from the mentioned ones were quantified in the lipidic extracts. It can be seen in Table 8 that for all the fatty acids quantified, there are very low values, but comparing these, a higher content can be seen in the ultrasound extract than in the macerated one, the difference being almost double, as is the case for the lipid content.

Similar results were obtained by Dulf et al. (2012) [65] and by Buya et al. (2012) [66], who studied a Mongolian variety of sea buckthorn.

Regarding the residues, the differences among R-SBM, R-SBUS, and SB are not significant, making it valuable for further valorization.

In order to explore the relationship between antioxidant activity and TPC and TFC, Pearson’s correlation coefficient (r) was calculated, and the results are shown in Table 9. The high antioxidant activity of the samples may be explained by the strong correlation between TPC and antioxidant value (r = 0.9881 and 0.9994). The Pearson’s correlation coefficients between TFC (mg RE/g) and antioxidant activity values (r = 0.9166 and 0.9729) are lower than those of TPC, values that are consistent with previous literature reports [52,53,54]. On the other hand, those between TFC (mg QE/g) and antioxidant activity values (r = 0.9953 and 0.9955) are lower in the case of CUPRAC but higher in case of FRAP II.

## 4. Conclusions

The valorization of by-product and wastes from the food industry is both a challenge and a critical desideratum in the current global context. Sea-buckthorn is a plant with important pharmaceutical effects conferred by the polyphenols, flavonoids, vitamins, minerals, and fatty acids content, also by its antioxidant activity. In this study, it was proved that the sea-buckthorn by-product resulting from the juice extraction can be further valorized by different extraction and concentration processes. From the obtained results, it can be seen that both maceration and ultrasound extractions are promising processes for the recovery of important phytochemical components, including polyphenols and flavonoids, and essential nutrients such as minerals, lipids, and fatty acids.

The ultrasound extraction process showed better results for all the studied parameters. The polyphenol content was 53% higher for E-SBUS than for E-SBM and 7.1 times higher than the sea-buckthorn by-product, and the flavonoid content was approximately 3 times higher for both extracts expressed in rutin and 2.5 times for the E-SBM and 4 times higher for E-SBUS expressed in quercetin. Additionally, the antioxidant activity showed similar results to the flavonoid content, the ultrasonication extraction process being more efficient than maceration and the antioxidant activity for E-SBUS being almost 2 times higher expressed in Trolox equivalents. The antioxidant activity of E-SBUS is approximately four times higher than the sea-buckthorn by-product.

Even if from the mineral analysis there cannot be expressed a general conclusion for all the elements, it can be seen that both concentrated extracts have almost 25% of the RDI value of K and Mg, and the content of Zn, Mn, and Fe is significant.

The lipid content of E-SBUS and E-SBM are insignificant, and ethanol solutions are not a solvent with the ability to solubilize and extract fatty substances from the vegetable matrix, but the content in E-SBUS is almost double that of E-SBM. The same trend can be observed for the fatty acid contents.

Statistical analysis shows a good correlation between antioxidant activity and TPC, respectively TFC.

## Figures and Tables

**Figure 1 foods-11-00462-f001:**
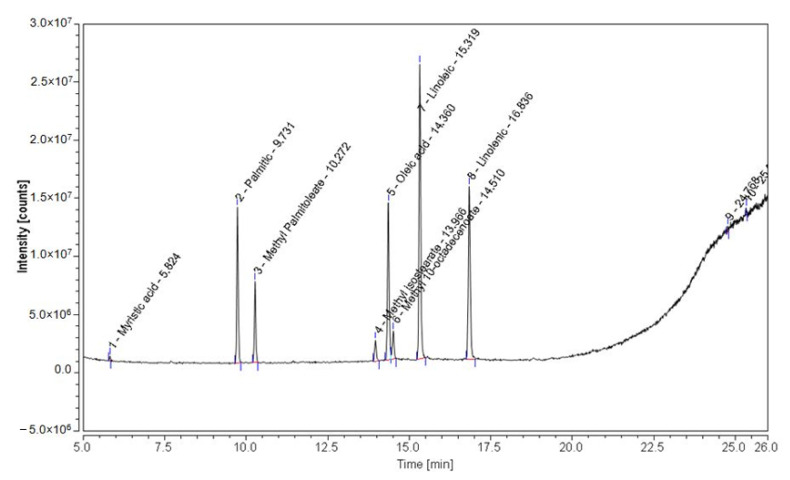
Chromatographic profile of fatty acids found in sea-buckthorn by-product, in order of their retention times: 1. myristic (RT 5.824), 2. palmitic (RT 9.731), 3. methyl palmitoleate (RT 10.272), 4. methyl isostearate (RT 13.966), 5. oleic acid (RT 14.360), 6. methyl 10-octadecenoate (RT 14.510), 7. linoleic (RT 15.319), 8. linolenic (RT 16.836).

**Table 1 foods-11-00462-t001:** Types of materials used in the experiments.

**SB**	Sea buckthorn by-product
**E-SBM**	Extract of sea buckthorn by-product obtained by maceration with recirculation and percolation
**E-SBUS**	Extract of sea buckthorn by-product obtained by ultrasound extraction
**R-SBM**	Residue after extraction of sea berry by-product by maceration with recirculation and percolation
**R-SBUS**	Residue after extraction of sea berry by-product by ultrasound extraction

**Table 2 foods-11-00462-t002:** Total polyphenols (TPC-mg GAE/g) content of samples.

Sample	TPC (mg GAE/g DM)	CV [%]
SB	32.55 ± 0.86 ^c^	2.64
E-SBM	105.16 ± 1.72 ^b^	1.63
E-SBUS	160.78 ± 7.21 ^a^	4.48
R-SBM	23.30 ± 0.41 ^d^	1.76
R-SBUS	22.57 ± 0.46 ^d^	2.03

Note: Values are the means of triplicate determinations. The results are presented as mean values ± standard deviation. Values followed by different letters in the same column are significantly different (*p* < 0.05). CV—Coefficient of variation.

**Table 3 foods-11-00462-t003:** Total flavonoid TFC (expressed in mg RE/g and mg QE/g) content of samples.

Sample	TFC (mg RE/g DM)	CV [%]	TFC (mg QE/g DM)	CV [%]
SB	3.60 ± 0.34 ^b^	9.44	50.52 ± 3.64 ^c^	7.20
E-SBM	10.42 ± 0.21 ^a^	2.01	129.57 ± 3.95 ^b^	3.04
E-SBUS	11.22 ± 1.89 ^a^	16.84	206.38 ± 21.68 ^a^	10.50
R-SBM	2.80 ± 0.34 ^b^	12.14	41.43 ± 4.37 ^c^	10.54
R-SBUS	2.99 ± 0.26 ^b^	8.69	42.95 ± 2.56 ^c^	5.96

Note: Values are the means of triplicate determinations. The results are presented as mean values ± standard deviation. Values followed by different letters in the same column are significantly different (*p* < 0.05); CV—Coefficient of variation.

**Table 4 foods-11-00462-t004:** Antioxidant activity (expressed in mg Trolox/g and mg/g Fe^2+^) content of samples.

Sample	Antioxidant Activity CUPRAC (mg Trolox/g DM)	CV [%]	Antioxidant Activity FRAP II (mg/g Fe^2+^ DM)	CV [%]
SB	88.99 ± 1.52 ^c^	1.70	101.36 ± 7.89 ^c^	7.78
E-SBM	229.38 ± 16.10 ^b^	7.01	287.93 ± 37.73 ^b^	13.10
E-SBUS	438.66 ± 16.21 ^a^	3.69	416.66 ± 60.94 ^a^	14.62
R-SBM	67.26 ± 3.07 ^c^	4.56	76.25 ± 15.63 ^c^	20.49
R-SBUS	64.65 ± 1.82 ^c^	2.81	66.84 ± 8.67 ^c^	12.97

Note: Each value represents a mean of three replicates. The results are expressed as mean ± standard deviation. Values followed by different letters in the same column are significantly different (*p* < 0.05).

**Table 5 foods-11-00462-t005:** Mineral content of samples.

Sample/ Determination (mg/100g DM)	Zn	Cu	Mn	Ca	Fe	Mg	Na	K
SB	4.51 ± 0.090 ^b^	1.23 ± 0.073 ^b^	3.51 ± 0.098 ^b^	40.15 ± 0.531 ^a^	3.6 ± 0.079 ^c^	60 ± 0.818 ^d^	40 ± 0.742 ^c^	450 ± 1.038 ^c^
E-SBM	3.63 ± 0.066 ^c^	<1 ± 0.00 ^c^	1.8 ± 0.072 ^d^	10.89 ± 0.314 ^c^	3.39 ± 0.055 ^d^	254 ± 1.48 ^a^	94 ± 0.703 ^b^	1208 ± 1.803 ^b^
E-SBUS	3.81 ± 0.081 ^c^	<1 ± 0.00 ^c^	1.6 ± 0.06 ^d^	13.11 ± 0.45 ^b^	3.9 ± 0.07 ^b,c^	183 ± 1.4 ^b^	105 ± 0.786 ^a^	1440 ± 2 ^a^
R-SBM	5.34 ± 0.094 ^a^	1.28 ± 0.092 ^b^	3.1 ± 0.09 ^c^	40.14 ± 0.522 ^a^	4.12 ± 0.157 ^a^	40 ± 0.665 ^e^	30 ± 0.52 ^d^	260 ± 0.916 ^d^
R-SBUS	5.41 ± 0.095 ^a^	1.5 ± 0.104 ^a^	3.82 ± 0.116 ^a^	40.16 ± 0.568 ^a^	3.7 ± 0.055 ^c^	50 ± 0.824 ^d^	30 ± 0.474 ^d^	240 ± 0.953 ^e^

Note: Each value represents a mean of three replicates. The results are expressed as mean ± standard deviation. Values followed by different letters in the same column are significantly different (*p* < 0.05).

**Table 6 foods-11-00462-t006:** The Daily Values of nutrients recommended per day(RDI) [63].

Constituents	RDI (FDA 2011) mg
Potassium	4700
Calcium	1000
Magnesium	400
Iron	18
Sodium	2400
Zinc	15
Manganese	4
Copper	2

Source: FDA http://www.fda.gov/nutritioneducation (accessed on 11 December 2021).

**Table 7 foods-11-00462-t007:** Total lipid content of samples.

Sample	Lipid Content (mg/g DM)	CV (%)
SB	8.96 ± 0.026 ^a^	0.29
E-SBM	0.06 ± 0.005 ^d^	8.33
E-SBUS	0.13 ± 0.010 ^e^	7.69
R-SBM	8.26 ± 0.015 ^c^	0.18
R-SBUS	8.37 ± 0.020 ^b^	0.23

Note: Each value represents a mean of three replicates. The results are expressed as mean ± standard deviation. Values followed by different letters in the same column are significantly different (*p* < 0.05).

**Table 8 foods-11-00462-t008:** The fatty acid contents of samples.

Sample/Determination (mg/g DM)	SB	E-SBM	E-SBUS	R-SBM	R-SBUS
Palmitic acid	0.935 ± 0.0283 ^a^	0.017 ± 0.0005 ^c^	0.023 ± 0.0049 ^c^	0.954 ± 0.0225 ^a^	0.906 ± 0.0094 ^a,b^
Methyl Palmitoleate	0.526 ± 0.0145^a^	0.0131 ± 0.0005 ^b^	0.0217 ± 0.0007 ^b^	0.538 ± 0.0061 ^a^	0.516 ± 0.0310 ^a^
Oleic	1.150 ± 0.0449 ^a^	0.0234 ± 0.0007 ^b^	0.0369 ± 0.0020 ^b^	1.205 ± 0.0267 ^a^	1.114 ± 0.0831 ^a^
Linoleic	2.255 ± 0.1255 ^a^	0.0302 ± 0.0020 ^b^	0.0517 ± 0.0015 ^b^	2.248 ± 0.1552 ^a^	2.082 ± 0.1303 ^a^
Linolenic	1.616 ± 0.2154 ^a^	0.0234 ± 0.0005 ^b^	0.0455 ± 0.0016 ^b^	1.625 ± 0.1010 ^a^	1.482 ± 0.1298 ^a^

Note: The results are expressed as mean ± standard deviation. Different letters in the same row indicate significant differences (*p* < 0.05).

**Table 9 foods-11-00462-t009:** Correlations among antioxidant activity and TPC and TFC.

	Antioxidant Activity CUPRAC (mg Trolox/g DM)	Antioxidant Activity FRAP II (mg/g Fe^2+^ DM)
AA CUPRAC (mg Trolox/g DM)	1	0.9835 **
AA FRAP II (mg/g Fe^2+^ DM)	0.9835 **	1
TPC (mg GAE/g)	0.9881 **	0.9994 ****
TFC (mg RE/g)	0.9166 *	0.9729 **
TFC (mg QE/g)	0.9953 ***	0.9955 ***

Note: **** Pearson correlation at *p* < 0.0001, *** Pearson correlation at *p* < 0.001, ** Pearson correlation at *p* < 0.01, * Pearson correlation at *p* < 0.05.

## Data Availability

Not applicable.

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
