# Peer review of "Reclaim and Valorization of Sea Buckthorn (Hippophae rhamnoides) By-Product: Antioxidant Activity and Chemical Characterization"

_foods, 2022, doi:10.3390/foods11030462_

Round 1

Reviewer 1 Report

The manuscript titled “Antioxidant and chemical characterization of sea buckthorn 2 (Hippophae rhamnoides) byproducts for further valorization in nutraceuticals” fits well with the Aims and Scope of FOODS and with the aims of the SI “ Ecofriendly Valorization of New Sources of Ingredients for Food Fortifications”, and achieves results of practical use for the industry dimension by evaluating the most suitable extraction technology to valorize the sea berry by-product in the nutraceutical area. However, I’ve found quite confusing the experimental design of the M&M section and above all the definition of employed samples. This last aspect inevitably affects the R&D section, which needs also to be deepened with practical considerations on the relevance of the by-product and the effectiveness of employed extraction technologies based on the obtained data, also in view of the wide range of plant by-products already addressed in literature.

Last but not least, the English language needs mandatorily to be revised throughout the manuscript.

For all these reasons, in my opinion, major comments are as follows.

In the title I’d not talk of “sea buckthorn by-products” rather “sea buckthorn by-product”. Please be consistent also throughout the manuscript.

Also, I’d change the title as follows:

Reclaim and valorization of sea buckthorn (Hippophae rhamnoides) by-product: antioxidant activity and chemical characterization

Abstract:

Page 1, lines 16-17: Reformulate as follows: “The by-product resulting from the production of the sea-buckthorn (Hippophae rhamnoides) juice may be a functional food ingredient..”

In the phrase “great source of bioactive compounds” I’d use “valuable”, “precious” or “interesting” rather than “great”…

 Page 1, lines 20-23: Reformulate as follows: “Next, sea-buckthorn waste (SB sample), extracts (SB-ME and SB-USE samples) and the residues obtained from the extractions (SBM and SBUS samples) were characterized for the total polyphenols, and flavonoid content, antioxidant capacity, mineral contents and fatty acids profile.”

Page 1, line 25: “and approximately 45% higher…”

No mention to the residues from extractions was made. Please, add some considerations also about them.

Introduction

Page 1, lines 35-37: cite recent efforts in reclaiming and valorizing by-products from the agri-food chain. For example:

  • Albergamo, A., Salvo, A., Carabetta, S., Arrigo, S., Di Sanzo, R., Costa, R., ... & Russo, M. (2021). Development of an antioxidant formula based on peanut by‐products and effects on sensory properties and aroma stability of fortified peanut snacks during storage. Journal of the Science of Food and Agriculture101(2), 638-647.

  • Albergamo, A., Costa, R., Bartolomeo, G., Rando, R., Vadalà, R., Nava, V., ... & Dugo, G. (2020). Grape water: reclaim and valorization of a by‐product from the industrial cryoconcentration of grape (Vitis vinifera) must. Journal of the Science of Food and Agriculture100(7), 2971-2981.

  • Costa, R., Albergamo, A., Arrigo, S., Gentile, F., & Dugo, G. (2019). Solid-phase microextraction-gas chromatography and ultra-high performance liquid chromatography applied to the characterization of lemon wax, a waste product from citrus industry. Journal of Chromatography A1603, 262-268.

  • Morales-Oyervides, L., Ruiz-Sánchez, J. P., Oliveira, J. C., Sousa-Gallagher, M. J., Morales-Martínez, T. K., Albergamo, A., ... & Montañez, J. (2020). Medium design from corncob hydrolyzate for pigment production by Talaromyces atroroseus GH2: Kinetics modeling and pigments characterization. Biochemical Engineering Journal161, 107698.

Page 1, line 44: polyunsaturated fatty acids…

Page 2, lines 56-57: “Thus, many efforts have been devoted to obtaining concentrated nutrients and antioxidant compounds from different by-products, such as grape pomace…” add and cite other examples of vegetable matrices.

At this stage of the manuscript, before describing the chosen extraction methods, I’d spend few lines on the relevance of finding the right technology for obtaining the highest yield of nutrients and bioactives from plant by-products…Cite also inherent literature.

Page 2, line 70: Correct as follows: Nikita Sawal et. al. [22]. Please be consistent throughout the manuscript.

Page 2, lines 92-95: Reformulate as follows: “Aim of this study was to reclaim the waste material resulting from the production of sea buckthorn juice and to evaluate the best extraction method between maceration and UAE, to obtain the highest yield of nutrients and bioactives from such by-product. To this purpose, both the extracts and the residues resulting from the extraction of seaberry by-product were evaluated in in terms of total polyphenols, flavonoids, antioxidant activity, mineral content, total lipid content and fatty acid composition.”

Page 3, lines 99-100: “Among other by-products, the study of the nutritional potential of sea buckthorn by-product may lead to new opportunities for solving food security(?), while increasing the planet's population and the need for high quality food resources.” However, pay attention to the term “food security”. In what sense do you use it? Clarify it…I’d mention here that the by-product may lead to new opportunities also for obtaining nutraceuticals and natural functional foods.

Page 3, lines 1032-104: I’d remove this period: “Also, consumers are more and more aware about the food quality from the nutritive point of view, so new food resources rich in bioactive compounds and are necessary to be exploited. Thus, these ingredients obtained from sea-buckthorn waste can be used successfully in obtaining food products with functional potential”. It’s basically a repetition of the previous sentence.

Material and methods:

Plant material and by-product production.

Please provide more information on fruit samples. When and where the fruits were collected? Info on the maturity stage? Amounts of fruits employed for the study?

Considering the by product, how did you obtain it? Please describe clearly the procedure employed for obtaining the juice and, therefore, the by-product. Also mention the amount of the by-product obtained after juice production.

2.2. Plant extracts            I’d talk of by-product extraction

On what basis did you choose these extraction protocols? Please specify it.

Overall, the description of samples employed in this study is quite confusing, as it seems you dealt with fruit samples and not by-product samples.

As a result, I propose to change the content of Table 1 as follows:

-SBà sea buckthorn by-product

-SB-ME changed into E-SBMà extract of sea buckthorn by-product obtained by maceration with recirculation and percolation

-SB-USE changed into E-SBUSà extract of sea buckthorn by-product obtained by ultrasound (and not ultrasonic) extraction

-SBM changed into R-SBM à residue after extraction of sea berry by-product by maceration with recirculation and percolation

-SBUS into R-SBUSà residue after extraction of sea berry by-product by ultrasound extraction

Please be consistent with these acronyms and relative descriptions throughout the manuscript.

Determination of total polyphenols (TPC)

Page lines 155-156:  “The sea-buckthorn byproducts were extracted…” I’d not talk of sea-buckthorn by-products…it would lead to confusion as, from now on, you dealt with extracts and residues from treated by-products. Hence, I’d simply write “The different samples were extracted…” or “SB, E-SBM, E-SBUS, R-SBM and R-SBUS samples were extracted…”. Please, be consistent throughout the manuscript.

2.11. Fatty acids quantificationà I’d talk of Fatty acid composition

From 2.9 to 2.11 paragraphs: I do not see cited protocols for the relative analysis.

Page 6, Line 254: “among samples.”

Results and Discussions

Please specify in every figure and table caption if Data are expressed on a dry weight or fresh weight basis.

I’d deepen every paragraph of this section with comparison of obtained data with data already present in literature on sea berry by-products or even other plant by-products, so that it can be derived the nutritional and nutraceutical relevance of sea berry by-product.

Reformulate as follows: 3.1. Total phenol content, flavonoid content, and antioxidant activities

Page 6, lines 276-279: “these results show that among the subsequent processing of the by-product, through different extraction processes, it can be further valorized with significant results, regarding the polyphenols content”. The period is not clear at all. Please reformulate it

Page 7, line 314:  When considering more than two items use “among” and not between. Please be consistent throughout the manuscript.

3.2. Mineral content analysis

What about the residue matrices?

Reviewer 2 Report

The authors described the results obtained from sea buckthorn byproducts differently extracted and concentrated from the nutraceutical point of view for further valorization and use in this field. The topic is quite interesting but, unfortunately, the manuscript presents many flaws.

Firstly, the manuscript lacks discussion, and no comparisons with literature data were done. The manuscript needs moderate English changes in the grammar and style; moreover, there are many typing errors (eg., Lines 38, 48, 70, 74, 76, 103,133, 377, etc.), wrong space (eg., Lines 30, 184, 229, 243, 248, 260, 262, etc.), superscript and subscript numbers, and punctuations need major revision. Moreover, ml or mL, M no g/L, by-product or byproduct, maybe quercetin instead quercitin, total polyphenols no total phenol, etc. should be uniform through the text. Further, the FRAP assay is usually expressed as uM or mg/g Fe2+ equivalents.
Many sentences are difficult to understand  (eg., Lines 118-119, 290-293, 306-316). Moreover, some wrong statements are reported, such as the fold increase values reported in lines 275 and 276 that are not relative to SBM and SBUS but to SB-ME and SB-USE, as well as polyphenols instead of flavonoids in lines 307 and 405.

Methods are not well-reported and important information is lacking (eg., sample extracts volumes used for each assay and the unit of measure for standards - mg, uM, etc). Moreover, the authors did not explain why used two different methods for flavonoids determination and recorded results at two different wavelengths.
Paragraph 2.3 should be shortened by merging reagents purchased from the same company and a section describing the sample extraction procedure should be included to avoid further repetitions.

All data must be shown only one time and any duplications need to be removed. To avoid unnecessary repetition, remove figures from the text.
Revise all table footnotes and remove superscript letters at the beginning.
Finally, regarding the sentence lines 361-363, have you used the same extraction procedure for all tested samples? If yes, this sentence is not appropriate. If not, you have to explain why used different extraction procedures.

Reviewer 3 Report

The manuscript is well written and structured. However, in my opinion, the authors should address some comments. 
My main concern is about why the authors used in used * in the determination of significant differences among treatments. I highly recommend changes to indicate those differences with letters. 
Additionally, I recommend to authors accomplish a correlation analysis between antioxidant activities and chemical characterization to identify the main compounds responsible for the bioactivity evaluated

Round 2

Reviewer 1 Report

Reviewer's suggestions have been accomplished.

Author Response

Minor English revisions were done.

Thank you!

Reviewer 2 Report

The paper has been significantly improved but the authors have not addressed all my comments.

Minor

Superscript and subscript numbers (e.g., fe2 +, Na2CO3, FeCl3, etc.), punctuation (e.g., Table 2,8,9: change the comma with a dot), and wrong spaces need further revision. Moreover, FRAP 2 or FRAP II, M or mol/l, etc., should be uniform throughout the text. 
The FRAP assay should be expressed as mg/g Fe2 + equivalents in the 2.8 section (and throughout the text) and correct as in the first submission "using a FeSO4*7H2O standard calibration curve".
The authors need to revise g/L and express it as M or mol/l.

Major
The volume of sample extracts used in each test has not yet been defined in the M&M section.

Author Response

Point 1. Superscript and subscript numbers (e.g., fe2 +, Na2CO3, FeCl3, etc.), punctuation (e.g., Table 2,8,9: change the comma with a dot), and wrong spaces need further revision. Moreover, FRAP 2 or FRAP II, M or mol/l, etc., should be uniform throughout the text. 
The FRAP assay should be expressed as mg/g Fe2 + equivalents in the 2.8 section (and throughout the text) and correct as in the first submission "using a FeSO4*7H2O standard calibration curve".
The authors need to revise g/L and express it as M or mol/l.

Response 1. Changes were made as suggested throughout the manuscript.

Point 2. The volume of sample extracts used in each test has not yet been defined in the M&M section.

Response 2. The following information's were added at the M&M section for each determination:

2.4. Determination of total polyphenols (TPC)

The extraction volumes used were: 0.05 ml for SB, R-SBM and R-SBUS, respectively of 0.015 ml for E-SBM and 0.03 ml for E-SBUS.

2.5. Determination of total flavonoid (TFC) as rutin equivalent

The extraction volumes used were: 5 ml for SB, R-SBM, R-SBUS, E-SBM and 4 ml for E-SBUS. 

2.6. Determination of total flavonoid (TFC) as quercetin equivalent

The extraction volumes used were: 0.5 ml for SB, R-SBM, R-SBUS, E-SBM and 0.35 ml for E-SBUS.

2.8. Antioxidant activity FRAP II assay

The extraction volumes used were: 0.015 ml for SB, 0.02 ml for E-SBM, E-SBUS and R-SBUS and 0.025 ml for R-SBM sample. 

Thank you!

Reviewer 3 Report

The authors have addressed all my comments for this paper and answered the questions. The paper has been significantly improved. I have no additional comments. Overall, the manuscript is well written and the reported results are of valuable interests to readers.

Author Response

(The authors gave the same response as above.)
